# PROVING THE STRONG LOTTERY TICKET HYPOTHESIS FOR CONVOLUTIONAL NEURAL NETWORKS

**Arthur C. W. da Cunha & Emanuele Natale**
Inria Sophia Antipolis
Sophia Antipolis, France
`{arthur.carvalho-walraven-da-cunha,emanuele.natale}@inria.fr`

**Laurent Viennot**
Inria Paris, IRIF
Paris, France
`laurent.viennot@inria.fr`

## ABSTRACT

The lottery ticket hypothesis states that a randomly-initialized neural network contains a small subnetwork which, when trained in isolation, can compete with the performance of the original network. Recent theoretical works proved an even stronger version: every sufficiently overparameterized (dense) neural network contains a subnetwork that, even without training, achieves accuracy comparable to that of the trained large network. These works left as an open problem to extend the result to convolutional neural networks (CNNs). In this work we provide such generalization by showing that, with high probability, it is possible to approximate any CNN by pruning a random CNN whose size is larger by a logarithmic factor.

## 1 INTRODUCTION

Many impressive successes in machine learning are reached through neural network architectures with a huge number of trainable parameters. Consequently, substantial research in the field aims at reducing the size of such networks while maintaining good accuracy; e.g., for deployment in resource constrained devices (Yang et al., 2017).

A major empirical fact of such endeavour is the contrast between the initial model overparametrization, which appears necessary for effective training, and the extent to which the size of the resulting model can be reduced through compression techniques. Among the latter, *pruning methods* appear as a mature and efficient way of achieving significant compression, often without incurring any accuracy loss (Blalock et al., 2020). Recently, the aforementioned contrast between the initial and final number of parameters has been addressed by the *lottery ticket hypothesis* (Frankle & Carbin, 2019), or LTH for short, which states that any randomly initialized network contains *lottery tickets*; that is, sparse subnetworks that can be trained just once and reach the performance of the fully-trained original network. This hypothesis was first verified experimentally, leveraging pruning methods to identify the lottery tickets (Frankle & Carbin, 2019; Lee et al., 2019).

Ramanujan et al. (2020) then proposed a stronger version of the hypothesis, named *strong lottery ticket hypothesis (SLTH)* by Pensia et al. (2020): it stipulates that a network with random weights contains, with high probability, sub-networks that can approximate any given sufficiently-smaller neural network. In other words, a sufficiently large and randomly initialized network that can be successfully trained for a task, could instead be suitably pruned to obtain a network that, even without training, achieves good accuracy. Experimental support for this stronger version were reported by Ramanujan et al. (2020); Zhou et al. (2019); Wang et al. (2020), which find lottery tickets in a range of architectures, including convolutional neural networks (CNNs). A first rigorous proof of the SLTH was given by Malach et al. (2020) for the case of dense networks (i.e., consisting of fully connected layers). Pensia et al. (2020) and Orseau et al. (2020) successively improved this result by showing that logarithmic over-parametrization is sufficient. Their results are also restricted to dense networks and they leave as an open problem to extend it to CNNs.

**Our contributions.** We extend and complete the proof of the SLTH (and thus, also, of the LTH), for classical network architectures which can combine convolutional and fully connected layers. More precisely, we prove that any CNN with given weights can be approximated by pruning a CNN with random weights (random CNN for short), with the latter being larger than the former by a logarithmic factor. We also provide basic experiments showing that starting from a random CNN which is roughly 30 times larger than LeNet5, it is possible to compute in few hours a pruning mask that allows to approximate the trained convolutional layers of LeNet5 with relative error $10^{-3}$, even when ignoring some hypothesis of our theoretical result. Our theoretical analysis follows the approach of Malach et al. (2020) and make use of two layers to approximate one. We borrow from Pensia et al. (2020) the use of random subset sum (RSS) (Lueker, 1998) to approximate a given weight via the sum of a subset of a sample of random weights, and carefully design instances of RSS via a combination of two convolutional layers. By controlling the error accumulated by each layer with Young's convolution inequality, we establish the following result.

> **Informal version of Theorem 1.** Given $\varepsilon, \delta > 0$, any CNN with $k$ parameters and $\ell$ layers, and kernels with $\ell_1$ norm at most 1, can be approximated within error $\varepsilon$ by pruning a random CNN with $O\left(k \log \frac{k\ell}{\min\{\varepsilon,\delta\}}\right)$ parameters and $2\ell$ layers with probability at least $1 - \delta$.

This result generalizes those by Pensia et al. (2020), Orseau et al. (2020), and Malach et al. (2020) as dense layers can be regarded as convolutional layers where kernel and input sizes match.

**Roadmap.** After discussing related work in the next section, we state our theoretical results alongside a high-level idea of the proofs. Successively, we report our experimental results. Finally, in Section 4, we provide detailed proofs of our statements.

## 1.1 RELATED WORK

Pruning methods are classical neural network compression strategies that date back to the 80s (LeCun et al., 1989b; Mozer & Smolensky, 1988). We recommend the recent survey Blalock et al. (2020) for an overview of the current state of research on these techniques.

As for the lottery ticket hypothesis, Lange (2020) summarizes the progress on the topic until the results by Malach et al. (2020). In the following we briefly mention works which are not discussed in Lange (2020). Cosentino et al. (2019) shows that lottery tickets can be adversarially trained, yielding sparse and robust neural networks. Soelen & Sheppard (2019) shows that lottery tickets are transferable, in the sense of showing remarkable accuracy for tasks other than the original one for which they have been found. Sabatelli et al. (2021) further shows that minimal retraining on a new task allows lottery tickets to often achieve better generalization than models trained ad-hoc for the task. Yu et al. (2020) empirically supports that the LTH holds also in the context of reinforcement learning and natural language processing. Fischer & Burkholz (2021) extends works on the SLTH to accommodate biases in practical settings. Diffenderfer & Kailkhura (2021) shows that lottery tickets are robust to extreme quantization of the weights. Aladago & Torresani (2021) provides a method to train networks where each initial weight is restricted to few possible random values. An extreme case of the latter is to share only a single (random) value among all weights, and focus the training solely on finding the best architecture (Gaier & Ha, 2019).

Our work also relates to recent papers investigating properties of random CNNs, such as Ulyanov et al. (2020) which observes that random CNNs already seem to capture some natural image statistics required for tasks such as de-noising and inpainting.

## 2 THEORETICAL RESULTS

We start by introducing some of our notation. The rest of it follows Goodfellow et al. (2016) with minor modifications, so we defer a full description to Section 4.1.

Given $n \in \mathbb{N}$, we denote the set $\{1, \ldots, n\}$ by $[n]$. The symbol $*$ represents the convolution operation, $\odot$ represents the element-wise (Hadamard) product, and $\sigma$ represents ReLU activation function. Finally, the notation $\|\cdot\|_1$ refers to the sum of the absolute values of each entry in a tensor while

$\|\cdot\|_{\max}$ denotes the maximum norm: the maximum among the absolute value of each entry. Those are akin to vector norms and should not be confused with operator norms.

We restrict our setting to convolutional neural networks $f \colon [0,1]^{D \times D \times c_0} \to \mathbb{R}^{D \times D \times c_\ell}$ of the form

$$f(\mathbf{X}) = \mathbf{K}^\ell * \sigma(\mathbf{K}^{\ell-1} * \cdots \sigma(\mathbf{K}^1 * \mathbf{X})),$$

where $\mathbf{K}^i \in \mathbb{R}^{d_i \times d_i \times c_{i-1} \times c_i}$, and the convolutions have no bias and are suitably padded with zeros. The restrictions on tensor sizes and the exclusion of bias terms[1] aim only to ease presentation.

Our initial goal is to approximate a convolution with a single kernel, as depicted in Figure 1, using convolutions with (pruned) random kernels. We achieve this by the means of the structure presented in Figure 2, using two convolutions with random tensors.

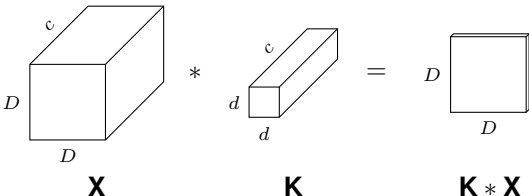

Figure 1: Schematics of the convolution between an input $\mathbf{X} \in \mathbb{R}^{D \times D \times c}$ and a kernel $\mathbf{K} \in \mathbb{R}^{d \times d \times c}$ resulting in a $D \times D$ matrix.

Lemma 1 asserts that, with high probability, we can prune this structure to approximate the output of a convolution with any given kernel as long as the amount of random kernels is large enough.

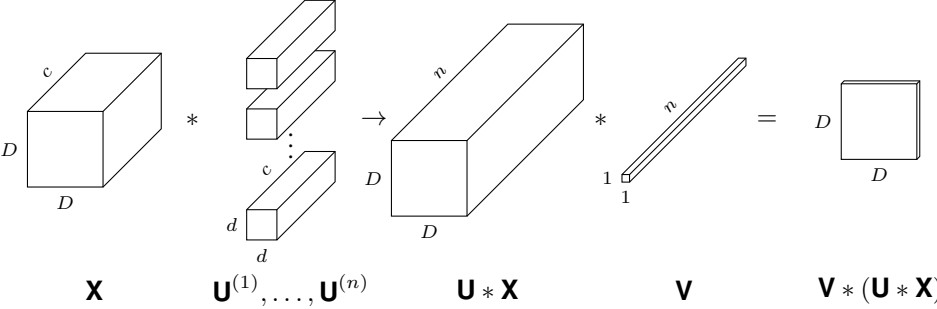

Figure 2: Schematics of the use of two convolutions to approximate the operation depicted in Figure 1. The elements of the set $\mathbf{U} = \{\mathbf{U}^{(1)}, \ldots, \mathbf{U}^{(n)}\}$ and $\mathbf{V}$ are random tensors. Notice that the intermediate tensor $\mathbf{U} * \mathbf{X}$ has size $D \times D \times n$ and yet the final output is a $D \times D$ matrix.

**Lemma 1** (Single kernel). *Let $D, d, c, n \in \mathbb{N}$, and $\varepsilon, C \in \mathbb{R}_{>0}$, where $n \geq C \log \frac{d^2 c}{\varepsilon}$, $\mathbf{U} \in \mathbb{R}^{d \times d \times c \times n}$, $\mathbf{V} \in \mathbb{R}^{1 \times 1 \times n \times 1}$, and $\mathbf{S} \in \{0,1\}^{\text{size}(\mathbf{U})}$, where the entries of $\mathbf{U}$ and $\mathbf{V}$ are i.i.d. $\text{Unif}([-1,1])$ random variables. Moreover, define the random CNN $g \colon [0,1]^{D \times D \times c} \to \mathbb{R}^{D \times D \times 1}$ and its pruned version $g_{\mathbf{S}}$ by*

$$g(\mathbf{X}) = \mathbf{V} * \sigma(\mathbf{U} * \mathbf{X}) \quad \text{and} \quad g_{\mathbf{S}}(\mathbf{X}) = \mathbf{V} * \sigma((\mathbf{U} \odot \mathbf{S}) * \mathbf{X}).$$

*Then, we can choose constant $C$ independently from other parameters so that, with probability at least $1 - \varepsilon$, for all $\mathbf{K} \in [-1,1]^{d \times d \times c \times 1}$ with $\|\mathbf{K}\|_1 \leq 1$, there exists a pruning mask $\mathbf{S}$ such that*

$$\sup_{\mathbf{X} \in [0,1]^{D \times D \times c}} \big\| \mathbf{K} * \mathbf{X} - g_{\mathbf{S}}(\mathbf{X}) \big\|_{\max} < \varepsilon.$$

*Proof idea.* We leverage the absence of negative entries in the input and an initial pruning of $\mathbf{U}$ to bypass the ReLU non-linearity. This allows us to virtually replace the operations in $g$ by a single

---

[1]If biases are present, the structures used in the proofs also puts them in a RSS configuration. Thus the results can be readily adapted by replacing the $d_i^2$ terms by $d_i^2 + 1$.

convolution with a random kernel obtained by combining $\mathbf{U}$ and $\mathbf{V}$. Each entry of this resulting kernel is the sum of $n$ random variables, where we can choose to include/exclude each term in the sum by choosing to keep/prune the relevant weights. We finish the proof by applying Theorem 2 to conclude that $n$ variables suffice to approximate all entries, simultaneously, with enough precision to ensure the thesis. $\square$

We now extend Lemma 1 to an entire layer. As before, a detailed proof is provided in Section 4.3.

**Lemma 2** (Convolutional Layer). *Let $D, d, c_0, c_1, n \in \mathbb{N}$, and $\varepsilon, C \in \mathbb{R}_{>0}$, where $n \geq Cc_1 \log \frac{d^2 c_0 c_1}{\varepsilon}$, $\mathbf{U} \in \mathbb{R}^{d \times d \times c_0 \times n}$, $\mathbf{V} \in \mathbb{R}^{1 \times 1 \times n \times c_1}$, $\mathbf{S} \in \{0,1\}^{\mathrm{size}(\mathbf{U})}$ and $\mathbf{T} \in \{0,1\}^{\mathrm{size}(\mathbf{V})}$, where the entries of $\mathbf{U}$ and $\mathbf{V}$ are i.i.d. $\mathrm{Unif}([-1,1])$ random variables. Finally, define the random CNN $g \colon [0,1]^{D \times D \times c_0} \to \mathbb{R}^{D \times D \times c_1}$ and its pruned version $g_{\mathbf{T},\mathbf{S}}(\mathbf{X})$ by*

$$g(\mathbf{X}) = \mathbf{V} * \sigma(\mathbf{U} * \mathbf{X}) \quad and \quad g_{\mathbf{T},\mathbf{S}}(\mathbf{X}) = (\mathbf{V} \odot \mathbf{T}) * \sigma((\mathbf{U} \odot \mathbf{S}) * \mathbf{X}).$$

*Then, we can choose constant $C$ independently from other parameters so that, with probability at least $1 - \varepsilon$, for all $\mathbf{K} \in [-1,1]^{d \times d \times c_0 \times c_1}$ with $\|\mathbf{K}\|_1 \leq 1$, there exist masks $\mathbf{S}$ and $\mathbf{T}$ such that*

$$\sup_{\mathbf{X} \in [0,1]^{D \times D \times c_0}} \|\mathbf{K} * \mathbf{X} - g_{\mathbf{T},\mathbf{S}}(\mathbf{X})\|_{\max} < \varepsilon.$$

*Proof Idea.* The lemma follows by applying Lemma 1 to each kernel independently, so that all of them are approximated by a factor of at most $\varepsilon/c_1$. Such approximation allows to apply the union bound so that the desired approximation holds simultaneously for all $c_1$ output kernels with probability at least $1 - \varepsilon$. $\square$

Next, we extend Lemma 2 from a single layer to the entire network, thus proving our main result. A detailed proof is given in Appendix B.

**Theorem 1** (Convolutional Network). *Let $D, d, c_0, \ell \in \mathbb{N}$, and $\varepsilon, C \in \mathbb{R}_{>0}$. For each $i \in [\ell]$, let $c_i, n_i \in \mathbb{N}$, where $n_i \geq Cc_i \log \frac{c_{i-1} c_i d_i^2 \ell}{\min\{\varepsilon, \delta\}}$, and $\mathbf{L}^{2i-1} \in \mathbb{R}^{d_i \times d_i \times c_{i-1} \times n_i}$, $\mathbf{L}^{2i} \in \mathbb{R}^{1 \times 1 \times n_i \times c_i}$, $\mathbf{S}^{2i-1} \in \{0,1\}^{\mathrm{size}(\mathbf{L}^{2i-1})}$, $\mathbf{S}^{2i} \in \{0,1\}^{\mathrm{size}(\mathbf{L}^{2i})}$, where the entries of $\mathbf{L}^1, \ldots, \mathbf{L}^{2\ell}$ are i.i.d. $\mathrm{Unif}([-1,1])$ random variables and define the random $2\ell$-layer CNN $g \colon [0,1]^{D \times D \times c_0} \to \mathbb{R}^{D \times D \times c_\ell}$ and its pruned version $g_{\mathbf{S}^1, \ldots, \mathbf{S}^{2\ell}}(\mathbf{X})$ by*

$$g(\mathbf{X}) = \mathbf{L}^{2\ell} * \sigma(\cdots \sigma(\mathbf{L}^1 * \mathbf{X})) \quad and \quad g_{\mathbf{S}^1, \ldots, \mathbf{S}^{2\ell}}(\mathbf{X}) = (\mathbf{L}^{2\ell} \odot \mathbf{S}^{2\ell}) * \sigma[\cdots \sigma[(\mathbf{L}^1 \odot \mathbf{S}^1) * \mathbf{X}]].$$

*Finally, let $\mathcal{F}$ be the class of functions from $[0,1]^{D \times D \times c_0}$ to $\mathbb{R}^{D \times D \times c_\ell}$ such that, for each $f \in \mathcal{F}$*

$$f(\mathbf{X}) = \mathbf{K}^\ell * \sigma(\mathbf{K}^{\ell-1} * \cdots \sigma(\mathbf{K}^1 * \mathbf{X})),$$

*where, for each $i \in [\ell]$, $\mathbf{K}^i \in [-1,1]^{d_i \times d_i \times c_{i-1} \times c_i}$ and $\|\mathbf{K}^i\|_1 \leq 1$.*

*Then, we can choose constant $C$ independently from other parameters so that, with probability at least $1 - \delta$, the following holds for every $f \in \mathcal{F}$:*

$$\inf_{\forall i \in [2\ell], \, \mathbf{S}^i \in \{0,1\}^{\mathrm{size}(\mathbf{L}^i)}} \sup_{\mathbf{X} \in [0,1]^{D \times D \times c_0}} \left\| f(\mathbf{X}) - g_{\mathbf{S}^1, \ldots, \mathbf{S}^{2\ell}}(\mathbf{X}) \right\|_{\max} < \varepsilon.$$

*Proof Idea.* The proof leverages Lemma 2 in an analogous way to how the latter relied on Lemma 1; namely, we apply Lemma 2 by requiring an approximation factor that guarantees, with sufficient probability, that a suitable approximation is reached across all layers simultaneously. The latter requirement is responsible for the $\ell$ factor which appears in the logarithms of the dimensions of each random tensor $\mathbf{L}_i$. $\square$

## 2.1 DISCUSSION ON THEOREM 1

**Size analysis.** For each layer, we emulate a 4-D kernel $\mathbf{K}$ with size $d_i \times d_i \times c_{i-1} \times c_i$ with two 4-D kernels $\mathbf{U}$ and $\mathbf{V}$ with size $d_i \times d_i \times c_{i-1} \times n$ and $1 \times 1 \times n \times c_i$ respectively with $n \geq Cc_i \log \frac{c_{i-1} c_i d_i^2 \ell}{\min\{\varepsilon, \delta\}}$. Under the technical assumption $c_i = O(d_i^2 c_{i-1})$ for $i \in [\ell]$, the size of $\mathbf{V}$ is within a constant factor

of that of **U**, and the whole random network we prune has size $O(k \log \frac{k\ell}{\min\{\varepsilon, \delta\}})$, where $k$ is the size of the network we want to approximate. This technical assumption is met for all classical convolutional networks used for image processing with a reasonably small constant in the big $O$ notation. We come back on this assumption below.

**Limitations.** The properties of convolutional layers require stronger hypotheses in Theorem 1 when compared with the results for dense layers Malach et al. (2020) or Pensia et al. (2020). First, we require non-negative inputs for all layers, however, since the output of the ReLU function is never negative, this restriction is only relevant for the input of the first layer. The mentioned works avoid this restriction by exploiting the identity $a = \sigma(a) - \sigma(-a)$ to deal separately with the positive and negative entries. The fact that each entry of the output of a convolution is affected by potentially multiple input entries prevents us from employing a similar strategy. Nonetheless, we remark that, while this is a relevant theoretical indication of the challenges imposed by the operation of convolution, in practice the inclusion of biases suffices to easily convert any CNN with a domain including negative values into an equivalent CNN that takes only non-negative inputs. Finally, the possibly multidimensional entries of convolutions also motivates the restriction on the norm of the target weight tensors in terms of the 1-norm.

**Generalizations.** For the sake of simplicity, we state and prove Theorem 1 in a restricted setting. It is worth remarking a series of generalizations that can be obtained at the mere cost of making the proofs more technically involved. We defer a more complete treatment of such extensions to a journal version of this work. First, the proof could consider also other parameters, such as stride, padding, average pooling and other operations that can be seen as convolutions. Moreover, we could consider more general convolutions, not necessarily 2-D, operating on tensors of any sufficiently large dimension. In particular, it is not necessary to assume that **V** has size $1 \times 1 \times n \times c_i$ in the above analysis. Using a 5-D tensor with size $d \times d \times c_{i-1} \times (n/c_i) \times c_1$ for **U**, an appropriate convolution **U** $*$ **X** would result in a $D \times D \times (n/c_i) \times c_i$ tensor, and we could use a $1 \times 1 \times (n/c_i) \times c_i$ tensor for **V** without the need for the mask **T** by performing in parallel $c_i$ appropriate convolutions. Note that the size of **V** is then smaller than the size of **U**. The technical assumption used in the above size analysis is thus not necessary to guarantee that logarithmic over-parametrization is sufficient. Finally, observe that that our results generalize to any probability distribution for the weights that contains a $b$-scaled Unif($[-a, a]$) for some constant $a > 0$ (in the sense of Definition 2 in Appendix C), where the parameters $a$ and $b$ only impact the constants in the theorem.

## 3 EXPERIMENTS

As networks with higher parameter count tend to be more robust to noise, we stick to the small CNN architecture used by Pensia et al. (2020), namely, LeNet5 (LeCun et al., 1989a) with ReLU activations. We conduct our experiments by first training a the network to 98.99% test accuracy on MNIST dataset (Lecun et al., 1998). To avoid well-known limitations of the MNIST dataset (in particular its large number of zero entries), we also trained it on the Fashion-MNIST dataset (Xiao et al., 2017) to 89.12% test accuracy. We adopted Kaiming Uniform (He et al., 2015) for weight initialization, a batch size of 64 and trained for 50 epochs using ADAM optimizer (Kingma & Ba, 2015) with learning rate of 0.001, exponential decay of 0.9 and momentum estimate of 0.999, the default values in Flux.jl (Innes et al., 2018) machine learning library.

Once the network is trained we change its weights for a random subset sum approximation of them. More precisely, for each weight $w$ we sample $\boldsymbol{x}$ from Unif($[-1, 1]^n$) and use Gurobi optimization software (Gurobi Optimization, LLC, 2021) to solve the mixed-integer program

$$\min_{a_1, \ldots, a_n} \left| w - \sum_{i=1}^{n} a_i \cdot x_i \right| \qquad \text{s.t.} \quad a_i \in \{0, 1\} \quad \forall i \in [n],$$

where $n$ is the sample size. Solving this subset sum problem with $n = 30$ for the 2572 parameters in the convolutional layers of LetNet takes around 1 hour on 32 cores of a Intel® Xeon® Gold 6240 CPU @ 2.60GHz.

Figure 3 shows the accuracy of the approximation for different sample sizes. We start to obtain good approximations (error smaller than $10^{-2}$) from sample sizes around 15-20. Also, when comparing

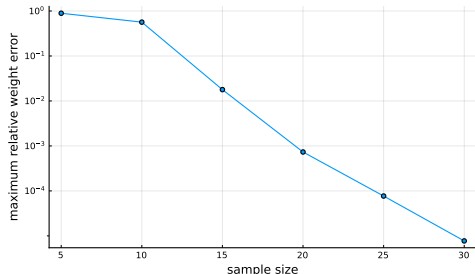 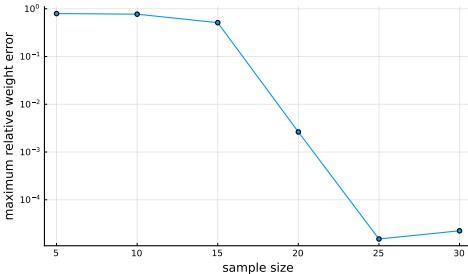

Figure 3: Relative error of random subset sum approximation of the convolution weights of a LeNet5 trained on MNIST (left) and on Fashion-MNIST (right). The error is given in logarithmic scale as the maximum distance between a weight and its approximation for different sample sizes.

to the weights obtained for MNIST and for Fashion-MNIST, we have better approximations for the smaller sample sizes for MNIST. We believe this is due to the fact that the training on Fashion-MNIST resulted in filters with larger weights (up to a factor 2, roughly), since a larger sample size is necessary to approximate a larger interval of values (see Theorem 2).

The high precision in the approximation of most weights leads to negligible change in the accuracy of the network. For this reason, we focus on studying the error at the output of the convolutional section of LeNet5, right before the flattening. Also, at this point the activation tensor has dimension $7 \times 7 \times 16$ as opposed to the vector of size 10 at the end of LeNet5.

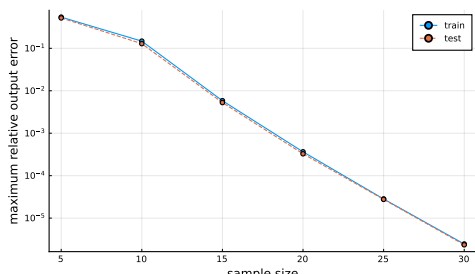 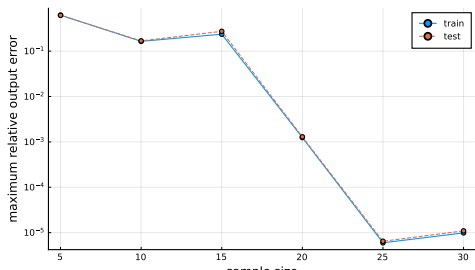

Figure 4: Maximum relative output error for the convolutional portion of LeNet5 trained on MNIST (left) and Fashion-MNIST (right) for different sample sizes. The maximum is computed over all images in the dataset.

Figure 4 shows the maximum relative error for all approximated outputs compared to original ones. The relative error of the output for an input image is computed as the maximum activation error divided by the maximum original activation (both maxima are taken over all $7 \times 7 \times 16$ activations). Once again, MNIST leads to better precision for the smaller sample sizes. This can be explained by the fact that weights are better approximated in that range with MNIST as seen in Figure 3. In both cases, we get a relative error close to $10^{-3}$ with sample size 20, and even better with larger sample sizes. Within the settings of Theorem 1, this corresponds to expanding the convolutional portion of the network by a factor of, roughly, 30 if we take into account kernel sizes and number of channels. This high precision is achieved even though the trained weights do not satisfy the norm restrictions of Theorem 1. Indeed, as we do not use any explicit regularization, the 1-norms of the kernels obtained are quite high (from 50 to 15000 roughly for both datasets).

## 4 TECHNICAL ANALYSIS

### 4.1 NOTATION AND CONVENTIONS

We employ the notation from Goodfellow et al. (2016) with minor adjustments. For self-containment, we detail below not only our adaptations but all the relevant notation inherited from the textbook.

We identify the type of the objects with the font used for their symbols. This applies to scalars (e.g. $x$), to vectors (e.g. $\boldsymbol{x}$) and its entries (e.g. $x_i$), to tensors (e.g. $\mathbf{X}$) and its entries (e.g. $X_{i,j,k}$), and to sets (e.g. $\mathbb{X}$). We denote slices of tensors by indexing it with colons. For example, the expression $\mathbf{X}_{:,:,i}$ represents a 2-D slice of a 3-D tensor. Finally, we refer to the axis of a 4-D tensors as *rows*, *columns*, *channels*, and *kernels* (a.k.a. filters), in this order.[2]

This work considers explicitly only 2-dimensional convolutions with multiple channels, multiple kernels and enough zero-padding to preserve the output size. However, as we discuss in Section 2.1, our results can be generalized in many ways.

**Definition 1** (Convolution). *Given tensors* $\mathbf{K} \in \mathbb{R}^{d \times d \times c}$ *and* $\mathbf{X} \in \mathbb{R}^{D \times D \times c}$, *the 2-dimensional discrete convolution between* $\mathbf{K}$ *and* $\mathbf{X}$ *is the* $D \times D$ *matrix with entries given by*

$$(\mathbf{K} * \mathbf{X})_{i,j} = \sum_{i',j' \in [d], k \in [c]} K_{i',j',k} \cdot X_{i-i'+1, j-j'+1, k} \quad \text{for } i, j \in [D].$$

*where* $\mathbf{X}$ *is suitably zero-padded. Similarly, if* $\mathbf{K} \in \mathbb{R}^{d \times d \times c_0 \times c_1}$ *is a 4-dimensional tensor, we define* $\mathbf{K} * \mathbf{X}$ *as the* $D \times D \times c_1$ *tensor with entries*

$$(\mathbf{K} * \mathbf{X})_{i,j,\ell} = \sum_{i',j' \in [d], k \in [c]} K_{i',j',k,\ell} \cdot X_{i-i'+1, j-j'+1, k} \quad \text{for } i, j \in [D], \ell \in [c_1].$$

The output of a convolution with a 4-D kernel can also be can be expressed in terms of the 3-D case using tensor slices. For instance, the equality above can be rephrased as

$$(\mathbf{K} * \mathbf{X})_{:,:,\ell} = \mathbf{K}_{:,:,:,\ell} * \mathbf{X} \quad \text{for } \ell \in [c_1].$$

### 4.2 SINGLE KERNEL APPROXIMATION (PROOF OF LEMMA 1)

Our first goal is to bypass the non-linearity so we can combine the two convolutions in $g(\mathbf{X}) = \mathbf{V} * \sigma(\mathbf{U} * \mathbf{X})$ into a single one. Given that the activation function under consideration is the ReLU, it suffices to ensure that its input has no negative entry. Hence, we prune all negative entries of $\mathbf{U}$, obtaining the tensor $\mathbf{U}^+ = \max\{\mathbf{0}, \mathbf{U}\}$, where the maximum is applied entry-wise. Since, by hypothesis, the entries of the input $\mathbf{X}$ are non-negative, it follows that the entries of the tensor $\mathbf{U}^+ * \mathbf{X}$ are also non-negative. Therefore,

$$\mathbf{V} * \sigma(\mathbf{U}^+ * \mathbf{X}) = \mathbf{V} * (\mathbf{U}^+ * \mathbf{X}). \tag{1}$$

We now look at the first convolution on the right side of Equation 1. By Definition (1), we have

$$\begin{aligned}
[\mathbf{V} * (\mathbf{U}^+ * \mathbf{X})]_{r,s,1} &= \sum_{t=1}^n V_{1,1,t,1} \cdot (\mathbf{U}^+ * \mathbf{X})_{r,s,t} \\
&= \sum_{t=1}^n V_{1,1,t,1} \cdot \left( \sum_{i,j \in [d], k \in [c]} U^+_{i,j,k,t} \cdot X_{r-i+1, s-j+1, k} \right) \\
&= \sum_{t=1}^n \sum_{i,j \in [d], k \in [c]} \left( V_{1,1,t,1} \cdot U^+_{i,j,k,t} \right) \cdot X_{r-i+1, s-j+1, k} \\
&= \sum_{i,j \in [d], k \in [c]} \left( \sum_{t=1}^n V_{1,1,t,1} \cdot U^+_{i,j,k,t} \right) \cdot X_{r-i+1, s-j+1, k}.
\end{aligned}$$

The equation above shows that performing $\mathbf{V} * (\mathbf{U}^+ * \mathbf{X})$ is equivalent to performing a single convolution between $\mathbf{X}$ and a tensor $\mathbf{L} \in \mathbb{R}^{d \times d \times c \times 1}$ whose coordinates are given by

$$L_{i,j,k,1} = \sum_{t=1}^n V_{1,1,t,1} \cdot U^+_{i,j,k,t}. \tag{2}$$

---

[2]Goodfellow et al. (2016) uses a different ordering.

This reveals a RSS configuration where we can choose to include/exclude each value $V_{1,1,t,1} \cdot U^+_{i,j,k,t}$ in the sum by choosing to keep/prune $U^+_{i,j,k,t}$. Since Equation 1 continues to hold after further pruning $\mathbf{U}^+$, we finish our proof by doing exactly that: we leverage Theorem 2 to ensure that, with high probability, we can solve this RSS problem for each entry of $\mathbf{L}$ to approximate the respective entry of $\mathbf{K}$.

To see that we can apply Theorem 2 in this setting, for $\varepsilon' > 0$, $i, j \in [d]$, and $k \in [c]$, denote by $E_{i,j,k,\varepsilon'}$ the event

$$\left\{ \forall z \in [-1, 1], \exists \mathbb{S} \subseteq [n] : \left| z - \sum_{t \in \mathbb{S}} V_{1,1,t,1} \cdot U^+_{i,j,k,t} \right| < \varepsilon' \right\}.$$

We now use the RSS result, Lueker (1998, Corollary 3.3) (Theorem 2 in Appendix C), to show that there exists constants $a, b$ such that

$$\mathbb{E}\left( \max_{z \in [-n/32, n/32]} \min_{\mathbb{S} \subseteq [n]} \left| z - \sum_{t \in \mathbb{S}} V_{1,1,t,1} \cdot U^+_{i,j,k,t} \right| \right) \le a e^{-bn}.$$

It is not hard to show that, since $(V_{1,1,t,1})_{1 \le t \le n}$ and $(U^+_{i,j,k,t})_{1 \le t \le n}$ are i.i.d. $\mathrm{Unif}([-1,1])$ random variables, then the value of the density of $(V_{1,1,t,1} \cdot U^+_{i,j,k,t})_{1 \le t \le n}$ is at least $\frac{\log 2}{2}$ on $[-1/2, 1/2]$, and, thus, it *contains* a $\frac{\log 2}{2}$-scaled $\mathrm{Unif}([-1/2, 1/2])$ (see Lemma 3 in Appendix C for details). In particular, setting $X = V_{1,1,t,1} \cdot U^+_{i,j,k,t}$, we have that $\mu_- = \mathbb{E}(\mathbf{1}_{X \le 0} X) \le -\frac{\log 2}{8} < -1/16$ and $\mu_+ = \mathbb{E}(\mathbf{1}_{X > 0} X) \ge \frac{\log 2}{8} > 1/16$. Therefore, we can apply Theorem 2 with $\xi = 1/32$: there exist constants $a, b > 0$ such that the expected value of the $[-n/32, n/32]$-subset-sum gap for $(V_{1,1,t,1} \cdot U^+_{i,j,k,t})_{1 \le t \le n}$ is at most $a e^{-bn}$. That is,

$$\mathbb{E}\left( \max_{z \in [-n/32, n/32]} \min_{\mathbb{S} \subseteq [n]} \left| z - \sum_{t \in \mathbb{S}} V_{1,1,t,1} \cdot U^+_{i,j,k,t} \right| \right) \le a e^{-bn}.$$

Assuming $n \ge 32$, Markov's inequality yields $\mathbb{P}(\overline{E_{i,j,k,\varepsilon'}}) \le \frac{a e^{-bn}}{\varepsilon'}$. Setting $\varepsilon' = \frac{\varepsilon}{d^2 c}$ and $C = \frac{2}{b} + \frac{\log a}{b}$, and supposing without loss of generality that $\varepsilon' < e^{-1}$, the condition $n \ge C \log \frac{1}{\varepsilon'}$ implies $bn \ge 2 \log \varepsilon' + \log a$ and $\frac{a e^{-bn}}{\varepsilon'} < \varepsilon'$, and we get

$$\mathbb{P}\left( E_{i,j,k,\frac{\varepsilon}{d^2 c}} \right) \ge 1 - \frac{\varepsilon}{d^2 c}.$$

Now define the simultaneous event $E_{\varepsilon'} = \bigcap_{i,j,k} E_{i,j,k,\varepsilon'}$. By a union bound over the inequality above for $i, j \in [d]$, $k \in [c]$, we have

$$\mathbb{P}\left( E_{\frac{\varepsilon}{d^2 c}} \right) \ge 1 - \varepsilon.$$

Finally, conditioning on $E_{\frac{\varepsilon}{d^2 c}}$, it holds that

$$\sup_{\mathbf{K} \in [0,1]^{d \times d \times 1 \times 1}} \inf_{\mathbf{S} \in \{0,1\}^{\mathrm{size}(\mathbf{U})}} \sup_{\mathbf{X} \in [0,1]^{D \times D \times 1}} \left\| \mathbf{K} * \mathbf{X} - \mathbf{V} * \sigma[(\mathbf{U} \odot \mathbf{S}) * \mathbf{X}] \right\|_{\max}$$

$$\overset{(a)}{=} \sup_{\mathbf{K}} \inf_{\mathbf{S}} \sup_{\mathbf{X}} \left\| \mathbf{K} * \mathbf{X} - \mathbf{V} * (\mathbf{U}^+ \odot \mathbf{S}) * \mathbf{X} \right\|_{\max}$$

$$\overset{(b)}{=} \sup_{\mathbf{K}} \inf_{\mathbf{S}} \sup_{\mathbf{X}} \left\| [\mathbf{K} - \mathbf{V} * (\mathbf{U}^+ \odot \mathbf{S})] * \mathbf{X} \right\|_{\max}$$

$$\overset{(c)}{\le} \sup_{\mathbf{K}} \inf_{\mathbf{S}} \sup_{\mathbf{X}} \left( \left\| \mathbf{K} - \mathbf{V} * (\mathbf{U}^+ \odot \mathbf{S}) \right\|_1 \cdot \left\| \mathbf{X} \right\|_{\max} \right)$$

$$\overset{(d)}{\le} \sup_{\mathbf{K}} \inf_{\mathbf{S}} \left\| \mathbf{K} - \mathbf{V} * (\mathbf{U}^+ \odot \mathbf{S}) \right\|_1$$

$$\overset{(e)}{\le} d^2 c \cdot \sup_{\mathbf{K}} \inf_{\mathbf{S}} \left\| \mathbf{K} - \mathbf{V} * (\mathbf{U}^+ \odot \mathbf{S}) \right\|_{\max}$$

$$\overset{(f)}{\le} d^2 c \frac{\varepsilon}{d^2 c} = \varepsilon,$$

where (a) follows from Equation 1, (b) from the distributivity of the convolution operation, (c) from Proposition 1, (d) from the fact that $\mathbf{X} \in [0,1]^{D \times D \times 1}$, (e) from the inequality $\|\boldsymbol{x}\|_1 \leq m\|\boldsymbol{x}\|_{\max}$ for $\boldsymbol{x} \in \mathbb{R}^m$, and (f) from the definition of $E_{\frac{\varepsilon}{cd^2}}$.

## 4.3 Convolutional Layer Approximation (Proof of Lemma 2)

The general goal of this argument is to choose binary masks $\mathbf{T}$ and $\mathbf{S}$ so that $(\mathbf{V} \odot \mathbf{T}) * \sigma[(\mathbf{U} \odot \mathbf{S}) * \mathbf{X}]$ is a sufficiently close approximation of $\mathbf{K} * \mathbf{X}$.

For $\ell \in [c_1]$ let $\mathbf{K}^{(\ell)}$ be $\mathbf{K}$'s $\ell$-th kernel. That is,

$$\mathbf{K}^{(\ell)} = \mathbf{K}_{:,:,:,\ell}.$$

Notice that $\mathbf{K} * \mathbf{X}$ is the concatenation along the third dimension of each $\mathbf{K}^{(\ell)} * \mathbf{X}$, i.e., for $\ell \in [c_1]$, we have $(\mathbf{K} * \mathbf{X})_{:,:,\ell} = \mathbf{K}^{(\ell)} * \mathbf{X}$.

We fix $\mathbf{T}$ a priori to be the block diagonal matrix $\mathbf{B}$ with entries given by $B_{1,1,t,\ell} = \mathbf{1}_{(\ell-1)n' < t \leq \ell n'}$ for $t \in [n], \ell \in [c_1]$, where $n' = n/c_1$. In the rest of the proof, we show how to choose $\mathbf{S}$, based on $\mathbf{U}$ and $\mathbf{V}$, in order to approximate the kernels $\mathbf{K}^{(\ell)}$.

We perform the approximation of each $\mathbf{K}^{(\ell)}$ using different sections of the tensors. To this end, for $\ell \in [c_1]$, let

$$\mathbf{U}^\ell = \mathbf{U}_{:,:,:,(\ell-1)n' < t \leq \ell n'}, \qquad \mathbf{S}^\ell = \mathbf{S}_{:,:,:,(\ell-1)n' < t \leq \ell n'}, \quad \text{and} \quad \mathbf{V}^\ell = \mathbf{V}_{:,:,(\ell-1)n' < t \leq \ell n',:}.$$

As we did in the proof of Lemma 1, we perform an initial pruning on $\mathbf{U}$ by restricting $\mathbf{S}$ to the space of masks that prune all of its negative entries. This allows us to ignore the ReLU activation and conclude that

$$
\begin{aligned}
&\left( (\mathbf{V} \odot \mathbf{B}) * \sigma[(\mathbf{U} \odot \mathbf{S}) * \mathbf{X}] \right)_{r,s,\ell} \\
&= \sum_{(\ell-1)n' < t \leq \ell n'} V_{1,1,t,\ell} \sum_{i,j \in [d], k \in [c]} (\mathbf{U} \odot \mathbf{S})_{i,j,k,t} \cdot X_{r-i+1,s-j+1,k} \\
&= \left( \mathbf{V}^\ell * [(\mathbf{U}^\ell \odot \mathbf{S}^\ell) * \mathbf{X}] \right)_{r,s} \\
&= \left( \mathbf{V}^\ell * \sigma[(\mathbf{U}^\ell \odot \mathbf{S}^\ell) * \mathbf{X}] \right)_{r,s}.
\end{aligned}
$$

For $\ell \in [c_1]$ and $\varepsilon' > 0$, denote by $E_{\ell,\varepsilon'}$ the event

$$\left\{ \sup_{\mathbf{K}^{(\ell)} \in [-1,1]^{d \times d \times c_0 \times 1}} \inf_{\mathbf{S}^\ell \in \{0,1\}^{\text{size}(\mathbf{U}^\ell)}} \sup_{\mathbf{X} \in [0,1]^{D \times D \times c_0}} \left\| \mathbf{K}^{(\ell)} * \mathbf{X} - \mathbf{V}^\ell * \sigma[(\mathbf{U}^\ell \odot \mathbf{S}^\ell) * \mathbf{X}] \right\|_{\max} < \varepsilon' \right\}.$$

Consider the event $E_{\varepsilon/c_1} = \bigcap_\ell E_{\ell,\varepsilon/c_1}$. Since $n' = n/c_1 = C \log \frac{d^2 c_0}{\varepsilon/c_1}$, for each $\ell \in [c_1]$, Lemma 1 ensures that $\mathbb{P}(E_{\ell,\varepsilon/c_1}) \geq 1 - \varepsilon/c_1$, which implies $\mathbb{P}(E_{\varepsilon/c_1}) \geq 1 - \varepsilon$.

Finally, conditioning on $E_{\varepsilon/c_1}$ and using the fact that the output channels of a convolutional layer are calculated independently, we conclude

$$
\begin{aligned}
&\sup_{\substack{\mathbf{K} \in [-1,1]^{d \times d \times c_0 \times c_1} \\ \mathbf{T} \in \{0,1\}^{\text{size}(\mathbf{V})}}} \inf_{\mathbf{S} \in \{0,1\}^{\text{size}(\mathbf{U})}} \sup_{\mathbf{X} \in [0,1]^{D \times D \times c_0}} \left\| \mathbf{K} * \mathbf{X} - (\mathbf{V} \odot \mathbf{T}) * \sigma[(\mathbf{U} \odot \mathbf{S}) * \mathbf{X}] \right\|_{\max} \\
&\leq \sup_{\mathbf{K} \in [-1,1]^{d \times d \times c_0 \times c_1}} \inf_{\mathbf{S} \in \{0,1\}^{\text{size}(\mathbf{U})}} \sup_{\mathbf{X} \in [0,1]^{D \times D \times c_0}} \left\| \mathbf{K} * \mathbf{X} - (\mathbf{V} \odot \mathbf{B}) * \sigma[(\mathbf{U} \odot \mathbf{S}) * \mathbf{X}] \right\|_{\max} \\
&= \max_{\ell \in [c_1]} \sup_{\mathbf{K}^{(\ell)} \in [-1,1]^{d \times d \times c_0}} \inf_{\mathbf{S}^\ell \in \{0,1\}^{\text{size}(\mathbf{U}^\ell)}} \sup_{\mathbf{X} \in [0,1]^{D \times D \times c_0}} \left\| \mathbf{K}^{(\ell)} * \mathbf{X} - \mathbf{V}^\ell * \sigma[(\mathbf{U}^\ell \odot \mathbf{S}^\ell) * \mathbf{X}] \right\|_{\max} \\
&< \varepsilon.
\end{aligned}
$$

## 5    REPRODUCIBILITY STATEMENT

We made our best effort to describe in detail all the relevant parameters for Section 3. We also took care to initialize the random number generator not only for Julia but also for Gurobi, so that the experiments can be reproduced using the source code available at `https://github.com/ArthurWalraven/cnnslth`. The latter also include a BSON serialized version of our trained models and solutions to RSS instances, that can readily be imported with Julia.

### ACKNOWLEDGMENTS

This work has been supported by the AID INRIA-DGA agreement n°2019650072. The authors are grateful to the OPAL infrastructure from Université Côte d'Azur for providing resources and support.

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

## A    BOUND ON THE NORM OF A CONVOLUTION

The following proposition can be seen as a special case of Young's Convolution Inequality for functions over $\mathbb{Z}^n$ where the norms in the inequality are the $\ell_1$ and $\ell_\infty$ norms.

**Proposition 1** (Tensor Convolution Inequality). *Given $\mathbf{K} \in \mathbb{R}^{d \times d \times c_0 \times c_1}$ and $\mathbf{X} \in \mathbb{R}^{D \times D \times c_0}$, we have*

$$\|\mathbf{K} * \mathbf{X}\|_{\max} \leq \|\mathbf{K}\|_1 \cdot \|\mathbf{X}\|_{\max}.$$

*Proof.* We have

$$\|\mathbf{K} * \mathbf{X}\|_{\max} \leq \max_{i,j \in [D], \ell \in [c_1]} \sum_{i',j' \in [d], k \in [c]} \left| K_{i',j',k,\ell} X_{i-i'+1, j-j'+1, k} \right|$$

$$\leq \max_{i,j,\ell} \sum_{i',j',k} \left| K_{i',j',k,\ell} \right| \|\mathbf{X}\|_{\max}$$

$$\leq \max_{i,j,\ell} \left( \sum_{i',j',k} |K_{i',j',k,\ell}| \right) \|\mathbf{X}\|_{\max}$$

$$\leq \max_{i,j,\ell} \|\mathbf{K}\|_1 \|\mathbf{X}\|_{\max} = \|\mathbf{K}\|_1 \cdot \|\mathbf{X}\|_{\max}.$$

$\square$

## B    CNN APPROXIMATION (PROOF OF THEOREM 1)

Since Lemma 2 provide bounds in terms of the output of a layer, the study of the propagation of this error through the network is mostly independent of the layer type. Hence, the next proof follows the structure of Pensia et al. (2020, Theorem 1), where our Lemma 2 assumes the role of their Lemma 3, and where we leverage our Proposition 1 in order to address the problem of bounding the maximum norm of a convolution.

*Proof (of Theorem 1).* For the sake of brevity, in the proof we denote the max-norm simply by $\|\cdot\|$. Let $\mathbf{X}^i$ be the input of the $i$-th layer of the network $f$. Thus,

1. $\mathbf{X}^1 = \mathbf{X}$,

2. $\mathbf{X}^{i+1} = \sigma(\mathbf{K}^i * \mathbf{X}^i)$ for $1 \leq i \leq \ell - 1$ and

3. $f(\mathbf{X}) = \mathbf{K}^\ell * \mathbf{X}^\ell$.

By applying Lemma 2 to each layer, we choose masks $\mathbf{S}^{2i}$ and $\mathbf{S}^{2i-1}$ so that

$$\sup_{\mathbf{X}} \left\| \mathbf{K}^i * \mathbf{X} - (\mathbf{L}^{2i} \odot \mathbf{S}^{2i}) * \sigma[(\mathbf{L}^{2i-1} \odot \mathbf{S}^{2i-1}) * \mathbf{X}] \right\| < \frac{\varepsilon}{2\ell} \tag{3}$$

with probability at least $1 - \frac{\varepsilon}{2\ell}$.

Since the ReLU function is 1-Lipschitz with respect to the max norm, the above event implies the following for all $i \in [\ell - 1]$:

$$\sup_{\mathbf{X}} \left\| \sigma(\mathbf{K}^i * \mathbf{X}) - \sigma[(\mathbf{L}^{2i} \odot \mathbf{S}^{2i}) * \sigma[(\mathbf{L}^{2i-1} \odot \mathbf{S}^{2i-1}) * \mathbf{X}]] \right\| < \frac{\varepsilon}{2\ell}. \tag{4}$$

By a union bound, with probability $1 - \varepsilon$, Equation 3 and Equation 4 hold for all layers simultaneously. In the rest of the proof, we implicitly condition on the latter event.

For any fixed function $f$, let $g$ be the pruned network constructed layer-wise, by pruning with binary masks which satisfy Equation 3 and Equation 4. Let these pruned tensors be $\tilde{\mathbf{L}}^i$, and let $\tilde{\mathbf{X}}^i$ be the input to the $(2i - 1)$-th layer of $g$.

We note that $\tilde{\mathbf{X}}^i$ satisfies the the recurrence relation

1. $\tilde{\mathbf{X}}^1 = \mathbf{X}$,

2. $\tilde{\mathbf{X}}^{i+1} = \sigma(\tilde{\mathbf{L}}^{2i} * \sigma(\tilde{\mathbf{L}}^{2i-1} * \mathbf{X}^i))$ for $1 \leq i \leq \ell - 1$.

Because $\|\mathbf{X}\| \leq 1$, Equation 4 implies that $\|\tilde{\mathbf{X}}^i\| \leq \left(1 + \frac{\varepsilon}{2\ell}\right)^{i-1}$. To see this, note that Equation 4 implies, for $1 \leq i \leq \ell - 1$,

$$\left\| \frac{\sigma(\mathbf{K}^i * \tilde{\mathbf{X}}^i)}{\|\tilde{\mathbf{X}}^i\|} - \frac{\tilde{\mathbf{X}}^{i+1}}{\|\tilde{\mathbf{X}}^i\|} \right\| \leq \frac{\varepsilon}{2\ell},$$

thus

$$\|\sigma(\mathbf{K}^i * \tilde{\mathbf{X}}^i) - \tilde{\mathbf{X}}^{i+1}\| \leq \frac{\varepsilon}{2\ell}\|\tilde{\mathbf{X}}^i\|.$$

By the reverse triangle inequality, the last inequality implies

$$\begin{aligned}
\|\tilde{\mathbf{X}}^{i+1}\| &\leq \frac{\varepsilon}{2\ell}\|\tilde{\mathbf{X}}^i\| + \|\sigma(\mathbf{K}^i * \tilde{\mathbf{X}}^i)\| \\
&\leq \frac{\varepsilon}{2\ell}\|\tilde{\mathbf{X}}^i\| + \|\mathbf{K}^i * \tilde{\mathbf{X}}^i\| \\
&\leq \frac{\varepsilon}{2\ell}\|\tilde{\mathbf{X}}^i\| + \|\mathbf{K}^i\|_1 \cdot \|\tilde{\mathbf{X}}^i\| \\
&\leq \frac{\varepsilon}{2\ell}\|\tilde{\mathbf{X}}^i\| + \|\tilde{\mathbf{X}}^i\| \\
&\leq \left(1 + \frac{\varepsilon}{2\ell}\right)\|\tilde{\mathbf{X}}^i\|.
\end{aligned}$$

Applying this inequality recursively, we get that $\|\tilde{\mathbf{X}}^i\| \leq \left(1 + \frac{\varepsilon}{2\ell}\right)^{i-1}$ for $1 \leq i \leq \ell - 1$. This allows us to bound the error between $\mathbf{X}^i$ and $\tilde{\mathbf{X}}^i$. For $1 \leq i \leq \ell - 1$, we have

$$\begin{aligned}
\|\mathbf{X}^{i+1} - \tilde{\mathbf{X}}^{i+1}\| &= \|\sigma(\mathbf{K}^i * \mathbf{X}^i) - \sigma(\tilde{\mathbf{L}}^{2i} * \sigma(\tilde{\mathbf{L}}^{2i-1} * \tilde{\mathbf{X}}^i))\| \\
&\leq \|\sigma(\mathbf{K}^i * \mathbf{X}^i) - \sigma(\mathbf{K}^i * \tilde{\mathbf{X}}^i)\| + \|\sigma(\mathbf{K}^i * \tilde{\mathbf{X}}^i) - \sigma(\tilde{\mathbf{L}}^{2i} * \sigma(\tilde{\mathbf{L}}^{2i-1} * \tilde{\mathbf{X}}^i))\| \\
&\leq \|\mathbf{K}^i\|_1 \|\mathbf{X}^i - \tilde{\mathbf{X}}^i\| + \|\sigma(\mathbf{K}^i * \tilde{\mathbf{X}}^i) - \sigma(\tilde{\mathbf{L}}^{2i} * \sigma(\tilde{\mathbf{L}}^{2i-1} * \tilde{\mathbf{X}}^i))\| \\
&\leq \|\mathbf{X}^i - \tilde{\mathbf{X}}^i\| + \|\sigma(\mathbf{K}^i * \tilde{\mathbf{X}}^i) - \sigma(\tilde{\mathbf{L}}^{2i} * \sigma(\tilde{\mathbf{L}}^{2i-1} * \tilde{\mathbf{X}}^i))\| \\
&\leq \|\mathbf{X}^i - \tilde{\mathbf{X}}^i\| + \left(1 + \frac{\varepsilon}{2\ell}\right)^{i-1}\frac{\varepsilon}{2\ell},
\end{aligned} \tag{5}$$

where for the last inequality we use Equation 3. Unrolling Equation 5 we get

$$\|\mathbf{X}^\ell - \tilde{\mathbf{X}}^\ell\| \leq \sum_{i=1}^{\ell-1}\left(1 + \frac{\varepsilon}{2\ell}\right)^{i-1}\frac{\varepsilon}{2\ell}.$$

Finally, this last inequality leads, with probability at least $1 - \varepsilon$, to

$$
\begin{aligned}
\|f(\mathbf{X}) - g(\mathbf{X})\| &= \|\mathbf{K}^\ell * \mathbf{X}^\ell - \tilde{\mathbf{L}}^{2\ell} * \sigma(\tilde{\mathbf{L}}^{2\ell-1} * \tilde{\mathbf{X}}^\ell)\| \\
&\leq \|\mathbf{K}^\ell * \mathbf{X}^\ell - \mathbf{K}^\ell * \tilde{\mathbf{X}}^\ell\| + \|\mathbf{K}^\ell * \tilde{\mathbf{X}}^\ell - \tilde{\mathbf{L}}^{2\ell} * \sigma(\tilde{\mathbf{L}}^{2\ell-1} * \tilde{\mathbf{X}}^\ell)\| \\
&\leq \|\mathbf{K}^\ell\|_1 \|\mathbf{X}^\ell - \tilde{\mathbf{X}}^\ell\| + \|\mathbf{K}^\ell * \tilde{\mathbf{X}}^\ell - \tilde{\mathbf{L}}^{2\ell} * \sigma(\tilde{\mathbf{L}}^{2\ell-1} * \tilde{\mathbf{X}}^\ell)\| \\
&\leq \|\mathbf{X}^\ell - \tilde{\mathbf{X}}^\ell\| + \|\mathbf{K}^\ell * \tilde{\mathbf{X}}^\ell - \tilde{\mathbf{L}}^{2\ell} * \sigma(\tilde{\mathbf{L}}^{2\ell-1} * \tilde{\mathbf{X}}^\ell)\| \\
&\leq \|\mathbf{X}^\ell - \tilde{\mathbf{X}}^\ell\| + \left(1 + \frac{\varepsilon}{2\ell}\right)^{\ell-1} \frac{\varepsilon}{2\ell} \\
&\leq \left(\sum_{i=1}^{\ell-1} \left(1 + \frac{\varepsilon}{2\ell}\right)^{i-1} \frac{\varepsilon}{2\ell}\right) + \left(1 + \frac{\varepsilon}{2\ell}\right)^{\ell-1} \frac{\varepsilon}{2\ell} \\
&\leq \sum_{i=1}^{\ell} \left(1 + \frac{\varepsilon}{2\ell}\right)^{i-1} \frac{\varepsilon}{2\ell} \\
&= \left(1 + \frac{\varepsilon}{2\ell}\right)^{\ell} - 1 \\
&< e^{\varepsilon/2} - 1 \\
&< \varepsilon,
\end{aligned}
$$

where the last inequality holds because $\varepsilon < 1$.

Replacing $\varepsilon$ in this proof with $\min\{\varepsilon, \delta\}$ concludes the proof of the theorem. $\qquad\square$

## C  RANDOM SUBSET-SUM THEOREM

For the sake of completeness, in this section we recall a result by Lueker (1998) together with the necessary definitions.

**Definition 2.** *Given two positive constants $a$ and $b$, we say that a distribution with density $f$ contains a $b$-scaled $\mathrm{Unif}([-a,a])$ distribution if for each $x \in [-a,a]$ it holds $f(x) \geq b$. We simply say that a distribution $F$ contains a uniform distribution if there exist positive constants $a$ and $b$ such that $F$ contains a $b$-scaled $\mathrm{Unif}([-a,a])$ distribution.*

The following is a weaker version of Corollary 1 in the Appendix of Pensia et al. (2020).

**Lemma 3.** *Let $X_1$ and $X_2$ be two independent random variables following a $\mathrm{Unif}([-1,1])$ distribution. Then $X_1 \cdot X_2$ contains a $\frac{\log 2}{2}$-scaled $\mathrm{Unif}([-\frac{1}{2}, \frac{1}{2}])$ distribution.*

We say that $z$ is $2\eta$-subsetsum-approximated with $S = \{X_1, \ldots, X_n\}$ if there exists a subset $I_z \subseteq [n]$ s.t. $|\sum_{i \in I_z} X_i - z| \leq 2\eta$.

**Definition 3.** *The $[a,b]$-subset-sum gap of $S = \{X_1, \ldots, X_n\}$ is the smallest value of $\eta$ such that each $z \in [a,b]$, can be $2\eta$-subsetsum-approximated with $S$.*

**Theorem 2** (Corollary 3.3 in Lueker (1998)). *Let $S = \{X_1, \ldots, X_n\}$ be $n$ i.i.d. bounded random variables and $\xi > 0$ any constant. Suppose that the distribution of $X_1$ contains a uniform distribution. Let $\mu_- = \mathbb{E}[\mathbf{1}_{X \leq 0} X]$, $\mu_+ = \mathbb{E}[\mathbf{1}_{X \geq 0} X]$, $\mu_{abs} = \mathbb{E}[|X|] = \mu_+ - \mu_-$. The expected value of the $[(\mu_- + \xi)n, (\mu_+ - \xi)n]$-subset-sum gap of $S$ is exponentially small with respect to $n$.*

