# OpenReview forum: "Proving the Lottery Ticket Hypothesis for Convolutional Neural Networks"
_ICLR.cc/2022/Conference — ICLR 2022 Poster_

### Official Review · Reviewer_dycZ · 2021-11-02

**Correctness:** 4
**Technical Novelty And Significance:** 3
**Empirical Novelty And Significance:** 2
**Recommendation:** 8
**Confidence:** 3

**Main Review:**

To begin with, I appreciate the efforts made by the authors that can help people to understand the mechanism behind the LTH. I am not an expert on theory nor random subset sum, but I can grasp some sense of the proof after reading the paper. However, there is still something that confuses me and I hope the authors can help me figure them out: Given a neural network, does the authors' theorem suggest that we need to use a pruned network of two times larger before we can approximate the real output of the original one? Theorem 1 seems to indicate that a pruned neural network of depth 2l approximates the output of f, whose depth is only l. In practice, such approximation can be done by the same architecture (same depth). There is a similar question for the number of parameters, whose bound seems to be controlled by the precision of approximation. I am not sure if this is nitpicking, but they do confuse me when I try to connect this theorem with LTH.

Minor problems:
- Figure 2, V * U * X should be V * (U * X) I suppose?
- In Lemma 1, g(X) was defined but never used afterward. The same for Lemma 2.
- Assumptions are too strong as indicated by the authors.


**Summary Of The Paper:**

The authors provide theoretical analysis on the lottery ticket hypothesis for convolutional neural networks. The techniques of the random subset sum are used to prove theorems. The authors start with introducing the case of a single convolutional kernel and then generalize to the case of the whole convolutional neural networks. By specifying multiple constraints and assumptions on the models' parameters and inputs, the error between the approximated and the true convolution results can be limited to a small number.


**Summary Of The Review:**

My score only indicates the initial opinion and I am willing to change my score if the authors can address my concern. Currently, I cannot clearly see the connection between the theorems and the LTH, so I would give a score of 5 at this time.

---

> ### Author Response · Authors · 2021-11-11
> **Response to the Review**
>
> We would like to thank the reviewer for the thoughtful comments. Your suggestions have been integrated into the paper and helped making it clearer. Please let us know if our answers address your concerns.
>
> #### 1. **Given a neural network, does the authors' theorem suggest that we need to use a pruned network of two times larger before we can approximate the real output of the original one? Theorem 1 seems to indicate that a pruned neural network of depth 2l approximates the output of f, whose depth is only l. In practice, such approximation can be done by the same architecture (same depth). There is a similar question for the number of parameters, whose bound seems to be controlled by the precision of approximation. I am not sure if this is nitpicking, but they do confuse me when I try to connect this theorem with LTH.**
>
> This is indeed a central point. The connection with the LTH can alternatively be understood as follows. For a given random CNN, we wish to know how well it can perform after pruning (without any training). Our theorem states that it can perform almost as well (if not better) as **the best** CNN among all those that are *sufficiently smaller* than the random CNN and whose weights can be suitably adjusted. In particular, such class includes classically trained networks. Thus, the result can be more broadly understood as a bridge between knowledge on networks found through standard methods and the emerging theory on pruning random networks.
>
> Notice that, in our results, *sufficiently smaller* means (a) half as many layers and (b) a number of parameters which is smaller by a logarithmic factor. With respect to (b), the logarithmic factor is indeed optimal (this can be seen via a packing argument, see e.g. [[PRN+]]).
> As for (a), we see reducing the number of layers as an interesting open problem.
>
> #### 2. **Figure 2, V * U * X should be V * (U * X) I suppose?**
>
> We thank the reviewer for noticing that. At that specific point, we are considering convolutions with a single kernel, so the operation in question is associative, but we still add the parenthesis in the text as a hint on the way we develop our argument. We forgot to add them in the figure and we have now fixed it.
>
> #### 3. **In Lemma 1, g(X) was defined but never used afterward. The same for Lemma 2.**
>
> We thank the reviewer for pointing out that the definition of $g$ was superfluous to the way the statements were written. We have now made explicit use of it and we think that the statements are now clearer.
>
>
> [PRN+]: https://arxiv.org/abs/2006.07990 "Pensia et al. (2020)"

---

> > ### Comment · Reviewer_dycZ · 2021-11-25
> > **Thanks for the response; Just one more clarification needed.**
> >
> > Thanks to the authors for answering my question. I can more understand the work and the challenges behind it. One thing I am curious about: what do the authors think is the main challenge to reduce the number of convolutional layers? From my perspective of view, I now more consider this work to be something on approximation theory, so I would follow the authors' thoughts and believe it is hard to do the approximation using the same number of layers. However, it would be better if the authors can provide more details on why it is hard, or even impossible with current tools, to prove the LTH (or do approximation). This is a question related to the application of authors' work, and it is not for evaluating its value.

---

> > > ### Author Response · Authors · 2021-11-29
> > > **Response to the Comment**
> > >
> > > We are glad to know our answer was helpful and we thank the reviewer for the discussion.
> > >
> > > We do believe it is a major challenge to reduce the number of layers while keeping the width within reasonable bounds.¹ This is an interesting direction for future work.
> > >
> > > Regarding proving the LTH, we remark the informality of its original statement. Depending on the interpretation, the hypothesis can be trivially proved or disproved.² Thus, works on the SLTH aim mostly at the principle behind the LTH: it claims that networks pruned at initialization can perform well once trained. Those works offer a way around the rather loose notion of training, by assuring the existence of subnetworks that perform well already before training.
> > >
> > > Nevertheless, we believe the SLTH implies a fair interpretation of the LTH.
> > > First, we emphasize that the target network in the SLTH does *not* correspond to the *lottery ticket*; in order to relate the SLTH to the LTH, the target network should be regarded as a mere tool of the analysis: from the point of view of the LTH, only the starting random network and its pruned subnetwork are visible. When the original random network for the LTH is assumed to be large enough, the class of target networks contains a network that achieves optimal loss for the task of interest, and the SLTH implies that the LTH holds.
> > >
> > > ¹With an exponential amount of neurons/filters, one can easily achieve an approximation with the same depth as the target network.
> > > ²A way to obtain the first would be to exploit the absence of a notion of sparsity in the statement, while the second could be derived from the lack of a lower bound on the size of the original network.

---

> > > > ### Comment · Reviewer_dycZ · 2021-11-29
> > > > **Thank you for your response**
> > > >
> > > > Thank you for your feedback. I have decided to raise my score after knowing that some of my questions are nitchpicking.

---

### Official Review · Reviewer_DkZV · 2021-11-02

**Correctness:** 4
**Technical Novelty And Significance:** 2
**Empirical Novelty And Significance:** Not applicable
**Recommendation:** 5
**Confidence:** 4

**Main Review:**

Strengths:
+ Convolutional layers are a frequently used building block in lottery ticket pruning. Related theoretical results have the potential to explain the empirical success in this area and derive conditions that enable pruning.
+ The paper is well written and explains all steps carefully.

Weaknesses:
- The proof only applies to positive inputs. This is a significant limitation, since neural ReLU networks can often be only successfully trained if the input is standardised and thus transformed to a range [-a,a]^n for a > 0.
- The parameter initialisation is unrealistic (iid from Unif([-1,1])) and does therefore not reflect the standard setting, in which pruning is conducted or neural networks are trained.
- The proof ideas are not novel. They main steps follow the work by Pensia et al. for fully connected layers with the simplification that only positive inputs are considered.
- The purpose of the experiments is unclear. It has been shown before that the subset sum approximation can be solved with roughly n=30 samples per parameter. This is not a new insight.
No neural networks are trained or actually pruned in the process. Why wouldn't the authors at least use the pruning algorithm edge-popup to show that they can approximate a target network?
Furthermore, why don't the authors report at least the test accuracy of the final lottery ticket to show that the approximation error of the complete pruned network is not too large and that their parameter-wise approximation error is not too high?
(The answer is probably that the proof does apply to a setting (i.e. parameter initialisation and positive inputs), in which edge-popup or any other gradient based algorithm would have a chance to find lottery tickets.)

Points of minor critique and open questions:
- How much larger is the initial large network than the target network in the experiments? The
- Why are the experiments restricted to only these small datasets (MNIST and Fashion-MNIST)? As no neural network needs to be trained, there is no strong computational overhead associated with experiments on more realistic image classification data.
- The proof is restricted to ReLU activation functions.
- The proof only applies to 2d convolutions. Extensions are reserved for a journal version.


**Summary Of The Paper:**

The lottery ticket hypothesis is proven for 2-dimensional convolutional network layers and positive inputs. It is shown that a given convolutional neural network can be approximated by pruning a randomly initialised larger network of two times the depth of the target network and width that is larger by a logarithmic factor.


**Summary Of The Review:**

Because of the lack of novel theoretical ideas and the fact that the proofs are limited to positive inputs and an unrealistic parameter initialisation approach, I recommend that the paper is not accepted for publication at ICLR at this stage.

---

> ### Author Response · Authors · 2021-11-11
> **Response to the Review (Part I)**
>
> We would like to thank the reviewer for their many questions, which we are of great help for improving the current version of our work.
>
> #### 1. **The proof ideas are not novel. They main steps follow the work by [[PRN+]] for fully connected layers with the simplification that only positive inputs are considered.**
>
> The main novelty of the paper is to prove SLTH on practical architectures mixing convolution and dense layers. We may better stress that it was unclear whether this could be achieved within logarithmic over-parameterization, as the structure of convolutions makes it more difficult to introduce an efficient mechanism for combining several weights into one such as subset sum. Concerning the technical novelty, although we reuse the subset sum trick of [[PRN+]], its introduction within convolutions required a different pruning approach and several tricks. To avoid repetition, we kindly refer the reviewer to our answer to Reviewer *ifdf* where we already detail this.
>
>
> #### 2. **The proof only applies to positive inputs. This is a significant limitation, since neural ReLU networks can often be only successfully trained if the input is standardised and thus transformed to a range [-a,a]^n for a > 0.**
>
> We think that assuming positive inputs is a mild restriction as the input can easily be shifted at the cost of changing the bias of each filter in the first layer. In fact, for an input $X \in [-1/2, 1/2]^{D \times D \times c}$ and a single filter $K$, we have
> $$K * X = K * (X + 1/2\cdot \mathbb{1}) - S/2 \cdot \mathbb{1}',$$
> where $S$ is the sum of values $K$'s entries and $\mathbb{1}$ and  $\mathbb{1}'$ are all-1 tensors with sizes $D \times D \times c$ and $D \times D$, respectively.
>
> More precisely, approximating a network $f$ with inputs in $[-1/2, 1/2]$ is equivalent to approximating a network $f'$ which takes shifted inputs in $[0,1]$ while being similar to $f$ with the exception of the biases in the first layer, which are modified appropriately.
>
> We would like to also recall that, regarding inclusion of biases, at the beginning of Section 2 we state that we develop most of our theory without mentioning them for the sole reason of easing the presentation, and we write in a footnote:
> > If biases are present, the structures used in the proofs also puts them in a RSS configuration. Thus the results can be readily adapted by replacing the $d_i^2$ terms by $d_i^2+ 1$.
>
>
> #### 3. **The parameter initialisation is unrealistic (iid from Unif([-1,1])) and does therefore not reflect the standard setting, in which pruning is conducted or neural networks are trained.**
>
> In order to keep the analysis as simple as possible, we tried to avoid any non-essential extension when stating our results, rather than directly providing them in their maximum generality. Regarding the distribution for the weight initialization, at the end of the *Generalization* paragraph in Section 2.1 we point out that our analysis holds for any distribution that *contains* a uniform one; at the best of our knowledge, this class covers all commonly used distributions used to initialize weights.
>
>
> *(Due to space restrictions, we have continued this answer in the comments below)*
>
>
> [PRN+]: https://arxiv.org/abs/2006.07990 "Pensia et al. (2020)"
> [OHR]: https://arxiv.org/abs/2006.12156 "Orseau et al. (2020)"

---

> > ### Author Response · Authors · 2021-11-11
> > **Response to the Review (Part II)**
> >
> > #### 4. **The purpose of the experiments is unclear. It has been shown before that the subset sum approximation can be solved with roughly n=30 samples per parameter. This is not a new insight. No neural networks are trained or actually pruned in the process. Why wouldn't the authors at least use the pruning algorithm edge-popup to show that they can approximate a target network? Furthermore, why don't the authors report at least the test accuracy of the final lottery ticket to show that the approximation error of the complete pruned network is not too large and that their parameter-wise approximation error is not too high?**
> >
> > Indeed, [[PRN+]] already indicates sample sizes that leads to a "good" approximation of the weights. Nonetheless, their experiments do not touch the convolutional layers and, since the weight sharing of convolutions makes them more sensitive to errors, their concept of "good" is weaker than ours. To put it in a different way, given a sample size, we should expect CNNs to have similar weight error when compared to dense networks, but we should still expect a higher output error.
> >
> > Regarding experiments around *edge-popup*, we believe they do not fit well the scope of the work. Our results are existence theorems and do aim to yield a practical algorithm to find lottery tickets. In fact, using the method underlying our proofs to this end would require prior knowledge of a smaller network that performs well on the desired task, which should defeat the point of a direct application.
> >
> > Finally, we mention that
> > > The high precision in the approximation of most weights leads to negligible change in the accuracy of the network.
> >
> > Because of the very nature of classification outputs (indicator vectors), accuracy measures can obfuscate output errors.
> > For instance, the vectors $(1,0)$ and $(0.51,0.5)$ are very different, but translate to the exact same Top-1 classification (first class).
> >
> > We hope these clarifications help to elucidate the empirical novelty of our approach when compared with prior works. We focused on providing experimental evidence for the thesis of our theorem outside the range of our hypothesis. That is, the purpose of our analysis is to provide empirical support to the SLTH for a wider class of networks.
> >
> >
> > #### 5. **How much larger is the initial large network than the target network in the experiments?**
> >
> > In our experiments we did not try to achieve the best compression ratio nor put any other constraint on the size of the subsets. As a consequence, the pruned random network turns out to be larger than the target network by a factor which is slightly less than half of the sample size. This reflects Gurobi's tendency towards the trivial solution (the empty subset) for small entries. We remark that experiments in [[PRN+]] exhibit this behaviour too. Studying the minimum subset size would require solving the problem by enumeration.
> > We think that obtaining results on the subset-sum problem with a better characterization of the subset size is an interesting direction for future work, since it could allow to better understand the sparsity of the random subnetwork.
> >
> >
> > #### 6. **Why are the experiments restricted to only these small datasets (MNIST and Fashion-MNIST)? As no neural network needs to be trained, there is no strong computational overhead associated with experiments on more realistic image classification data.**
> >
> > Working with larger networks requires to solve more subset sum instances. Leaving the computational cost aside, the problem is actually less challenging on larger datasets/architectures as those tend to be more robust to noise. As the reviewer indicates, this property is not sufficiently intuitive and we have made it explicit in the text.
> >
> >
> > #### 7. **The proof is restricted to ReLU activation functions.**
> >
> > This restriction can be relaxed a little by only employing ReLU at every second layer while the remaining layers could have any 1-Lipschitz activation. We believe that investigating the problem with different activation functions could be an interesting direction for future work.
> >
> >
> > #### 8. **The proof only applies to 2d convolutions. Extensions are reserved for a journal version.**
> >
> > As discussed in the "Generalizations" paragraph in Section 2.1, requiring to approximate 2D convolutions with 2D convolutions is more challenging than allowing for higher-dimension convolutions.
> > That is, while the 2D case does not imply the higher-dimension one, the former is stronger in the sense that generalizing from it to the latter does not pose technical challenges. Moreover, adopting 2D convolutions allows for a more accessible presentation, since 2D convolutions are more commonly used.
> >
> >
> > [PRN+]: https://arxiv.org/abs/2006.07990 "Pensia et al. (2020)"
> > [OHR]: https://arxiv.org/abs/2006.12156 "Orseau et al. (2020)"

---

> > > ### Comment · Reviewer_DkZV · 2021-11-18
> > > **Do the results provide relevant insights into LT existence?**
> > >
> > > I thanks the authors for the detailed clarifications.
> > >
> > > 4) I appreciate the pruning experiments by the authors that show that the subset sum errors are negligible for the classification performance of the pruned network.
> > > However, I disagree that edge-popup experiments do not fit the scope of the work for the following reason: Existence proofs should explain experiments and prove existence under realistic conditions. The random initial networks that are pruned need to have similar properties as the ones that are actually pruned by state-of-the-art algorithms, in this case edge-popup. If the authors would do the experiments, they would realize that they make unrealistic assumptions.
> > > a) Weights are initially not distributed iid from Unif([-1,1]) but from a scaled distribution Unif([-\sigma,\sigma]), where \sigma \propto 1/sqrt(width). This is necessary to prevent gradients from exploding. Standard pruning algorithms like edge-popup rely on these gradients.
> > > b) Obviously, input data can easily be shifted. But is the network trainable in this case?
> > > There is a reason why data is usually standardized before training.
> > > Btw, also [PRN+] presented experiments with edge-popup to verify that they make realistic assumptions.
> > >
> > > 5) I was actually wondering how large the large initial network is? Obviously, the pruned network needs to be larger than the target network if subset sum approximation is employed. I was just wondering if the large initial network has a size that is comparable with architectures that are trained in practice?

---

> > > > ### Author Response · Authors · 2021-11-21
> > > > **Response to Review**
> > > >
> > > > We thank the reviewer for the further input. Our results imply that sufficiently large random networks contain winning tickets that perform well even without training. As the existence of such subnetworks was not known to be general for CNNs, we do believe our work is a relevant contribution. On the other hand, the SLTH provides a formal interpretation of the LTH for which the notion of training becomes trivial. Thus, the theoretical works on the strong hypothesis do not provide much information on aspects of the LTH which concern training.
> > > >
> > > > 4. a) Our results can be easily generalized to cover a scaled down distribution at the cost of changing $C$. In any case, the positive-homogeneity of ReLU networks renders scaling issues less meaningful. For instance, classification tasks with normalization at output are invariant by scaling the weights.
> > > > b) The theorems also cover networks that could not be found by gradient based methods.
> > > > Experiment 2 in [PRN+] aims to evaluate the impacts of topological structures suggested by the proofs from that work. Since this requires pruning without training, they employ *edge-popup*. Their trials indicate that those topological changes do not perform better than a proportional increase in the width of the network, which was to be expected.
> > > >
> > > >
> > > > 5. Our experiments indicate that a network roughly 30 times wider suffices for good approximation.
> > > >
> > > >
> > > > [PRN+]: https://arxiv.org/abs/2006.07990 "Pensia et al. (2020)"

---

> > > > > ### Comment · Reviewer_DkZV · 2021-11-25
> > > > > **I acknowledge the author repsonse**
> > > > >
> > > > > I appreciate the response by the authors but dare to disagree that theoretical works on the strong hypothesis do not provide much information on aspects of the LTH which concern training, as they indicate under which conditions we can expect to be find lottery tickets. The question is whether the results are meaningful, as they prove the existence of lottery tickets under realistic conditions and thus in randomly initialized neural networks that are pruned in experiments.
> > > > >
> > > > > 4a) ReLU networks are invariant under rescaling only for non-zero biases and C would need to be huge in general.
> > > > > 4b) I agree that the theorems do not necessarily cover networks that can be found by gradient based methods. However, PRN+ show that the proof conditions are realistic (in terms of topology and initialization) with their edge-popup experiments.
> > > > >
> > > > > I agree with the authors, however, that the existence proof for CNNs is a relevant contribution but the paper would be stronger with more realistic assumptions.

---

> > > > > > ### Author Response · Authors · 2021-11-29
> > > > > > **Response to Review**
> > > > > >
> > > > > > We agree with the reviewer on the value of works on the SLTH and wish to briefly clarify our previous response.
> > > > > > The concept of lottery tickets, in general, refers to subnetworks that perform well after training. In the context of the SLTH, we focus on the subset of lottery tickets that perform well even before training. While we agree that those tickets could benefit from training, taking this into account in a rigorous treatment appears to be beyond the current techniques. Our former comment on the SLTH theory is referring to such restriction only.
> > > > > >
> > > > > > We also agree that the paper would be strong with weaker hypothesis. However we do not believe that experiments similar to those in [[PRN+]] would change the substance of our work. The relationship between theoretical results on the SLTH and gradient methods to find untrained subnetworks has many aspects that are beyond the scope of our paper. Some of them are addressed in the recent work [[FB]] which we plan to discuss in the camera-ready version.
> > > > > >
> > > > > > We thank the reviewer for the exchange. It allowed us to clarify many points about our work, and we hope that we also succeeded in explaining our thoughts on this last issue.
> > > > > >
> > > > > >
> > > > > > [PRN+]: https://arxiv.org/abs/2006.07990 "Pensia et al. (2020)"
> > > > > > [RWK+]: https://arxiv.org/pdf/1911.13299.pdf "Ramanujan et al. (2020)"
> > > > > > [FB]: https://arxiv.org/pdf/2110.11150.pdf "Fischer, Burkholz (2021)"

---

> > > > > > > ### Comment · Reviewer_DkZV · 2021-11-30
> > > > > > > **Response to authors**
> > > > > > >
> > > > > > > I thank the authors for sharing their thoughts and appreciate the discussion.
> > > > > > >
> > > > > > > I never suggested the integration of training in their existence proof. I only asked to consider the trainability of the large randomly initialized neural network to derive realistic network properties, which are common in contemporary work on lottery tickets.
> > > > > > > Experiments similar to those in [PRN+] would, of course, not change the substance of the authors work, which is the existence proof. I asked for those experiments because I suspect that the authors would not be able to perform them successfully, which would reveal that they prove the SLTH under unrealistic conditions.
> > > > > > > [FB] only consider fully connected feed forward networks, if I am not mistaken, not convolutional layers.

---

### Official Review · Reviewer_7cVT · 2021-11-03

**Correctness:** 4
**Technical Novelty And Significance:** 3
**Empirical Novelty And Significance:** 2
**Recommendation:** 8
**Confidence:** 4

**Main Review:**

This paper proves the STLH for convolutional neural networks (CNNs). The existing papers have all focused on proving the SLTH for fully connected networks, so this paper fills the gap by proving that SLTH also holds for convolutional networks. The overparameterization required is logarithmic in the approximation error and the network depth and number of parameters, which matches the existing results for fully connected networks by Pensia et al. (2020). The importance of this result is that this proves that (sufficiently large) randomly initialized CNNs can be 'trained' solely by pruning, without actually updating the weights of the CNN.

The proof technique and construction uses the Subset Sum approach, similar to Pensia et al. (2020), and overall the paper is well written.

I have the following questions:
1. It is not clear to me why the negative input situation cannot be handled similar to prior work. Even if the same kernel is applied to multiple inputs, as long as the kernel is well approximated, the outputs will also be well approximated, respectively for each input. Could the authors please provide some more explanation or an example?

2. The theorem and lemma statements say "... we can choose constant $C$ independently from other parameters ...". Does this mean that $C$ is a universal constant?

Typos:
In the Limitations paragraph on page 5, it should be $a=\sigma(a)-\sigma(-a)$ instead of $a=\sigma(a)-\sigma(a)$.

**Summary Of The Paper:**

The Strong Lottery Ticket Hypothesis (SLTH) says that any (sufficiently large) randomly initialized network can be pruned to obtain a network which performs well on a given task. This paper proves the Strong Lottery Ticket Hypothesis for convolutional neural networks by showing that given a target convolutional network, a logarithmically wider and twice deeper randomly initialized network can be pruned to approximate it.

**Summary Of The Review:**

The paper fills an important gap in existing research by proving that the Strong Lottery Ticket Hypothesis holds for Convolutional Neural Networks with only logarithmic overparameterization. However, the proof and construction is very similar to that in Malach et al. (2020) and Pensia et al. (2020), and hence the technical novelty in terms of proof techniques could be considered incremental.

---

> ### Author Response · Authors · 2021-11-11
> **Response to the Review**
>
> We thank the reviewer for their feedback and questions which provide us useful insights for improving the current version of our work. We are glad that the reviewer think that our results fill an "important gap" in existing research.
>
> #### 1. **It is not clear to me why the negative input situation cannot be handled similar to prior work. Even if the same kernel is applied to multiple inputs, as long as the kernel is well approximated, the outputs will also be well approximated, respectively for each input. Could the authors please provide some more explanation or an example?**
>
> A target network that takes negative inputs can be readily transformed into an equivalent one taking shifted (positive) inputs as explained in Answer 2 to Reviewer *DkZV*.
>
> In spite of that, it is true that a sufficiently good approximation of the kernel will provide a good approximation of the output. This would be enough for the case where the first of the two layers has a linear activation function. Yet, requiring the pruned network to have a similar structure as the target network implies adopting a ReLU activation for all layers, which demands additional care. In the scope of our work, the issue is that even if the tensor $V * U$ approximates the target tensor $K$, the tensor $U$ alone might not. To give a small example (with no padding), let $K = (1,-1)$, $U = (-1,1)$, and $V = (-1)$. We have $K = U * V$ and, nonetheless, for input $(0,-1,1)$, we have $K * (0,-1,1) = (1,-2)$, while $V * \sigma(U * (0,-1,1)) = (0,-2)$.
>
> The reviewer's point also gives us an opportunity to illustrate another situation where approaches from previous works would not lead to logarithmic bounds. The method introduced by [[MYSS]] handles the intermediate ReLU by leveraging the fact that the input is a scalar and as such, ignoring the trivial case of zero, can only be positive or negative. This allows for dealing with each case independently at the cost of doubling the amount of parameters: depending on the input sign, half of the output entries will be zero and calculations resume with the other half. For convolutions, however, since multidimensional inputs are possible, there is no clear analogous approach. A way of adapting the method would be, for each entry of the target tensor, to keep only the respective entry of the random tensor (pruning all the others) and then proceed like [[MYSS]] and prune it depending on the sign. While this would provide the desired properties, it requires a polynomial amount of random filters.
>
>
> #### 2. **The theorem and lemma statements say "... we can choose constant independently from other parameters ...". Does this mean that is a universal constant?**
>
> We confirm that the constant $C$ is universal. In our statements we keep it separate from other parameters (avoiding to absorb other constants), so that dependencies are explicit in the provided bound on $n$.  For example, the size of the input image $D$ is a free parameter, since the approximation of each kernel holds regardless of it.
>
> #### 3. **The proof and construction is very similar to that in [[MYSS]] and [[PRN+]], and hence the technical novelty in terms of proof techniques could be considered incremental.**
>
> This point has also been raised by Reviewer *ifdf*; to avoid repetition, we kindly defer the reviewer to the answer we provide in the comment to Reviewer *ifdf*'s feedback.
>
> [MYSS]: https://arxiv.org/abs/2002.00585 "Malach et al. (2020)"
> [PRN+]: https://arxiv.org/abs/2006.07990 "Pensia et al. (2020)"

---

### Official Review · Reviewer_ifdf · 2021-11-09

**Correctness:** 4
**Technical Novelty And Significance:** 3
**Empirical Novelty And Significance:** Not applicable
**Recommendation:** 6
**Confidence:** 2

**Main Review:**

This paper is an extension of previous theoretical results on SLTH from fully connected layers to convolutional layers. The proving technique is similar to that used in Malach et al. (2020) and Pensia et al. (2020). While I am a LTH practitioner, I am not very familiar with LTH and SLTH theory and hence I am not sure how significant the technical contribution it is to extend the theoretical analysis from Malach et al. (2020) and Pensia et al. (2020) and whether it suffices to reach the bar of this venue.

The paper is well written and easy to follow. I roughly went through the mathematical proofs and found no issue. The main theorem is well supported by the empirical experiments.

**Summary Of The Paper:**

This paper proves for the convolutional neural networks (CNNs) the strong Lottery Ticket Hypothesis, which previously is brought out based on empirical findings that a substantially overparameterized random neural network (with fully connected layers or convolutional layers) can be pruned to match the performance of a well-trained smaller network, even without training the remaining parameters. The main theorem states that a CNN with any given weights can be approximated well with high probability with a larger CNN with a doubled number of layers and a logarithmic number of parameters, and a proper pruning mask on random weights.

**Summary Of The Review:**

Overall, this work fills the gap between the empirical success of SLTH in CNNs and the lack of theoretical interpretation. However, the review is not sure if the technical contribution is significant enough compared to previous theoretical work on neural networks with fully connected layers.

---

> ### Author Response · Authors · 2021-11-11
> **Response to the Review**
>
> We would like to thank the reviewer for the feedback. We are glad that the reviewer agrees that our results fill a gap in existing research. The main issue raised concerns the technical novelty of the paper.
>
> We realize that our use of subset sum in convolution networks might seems natural, and that we should have better highlighted the difficulties behind it. Indeed, it requires to combine several non-trivial tricks, which is probably the reason why [[PRN+]] left such a generalization as an (interesting) open problem. Although we briefly mention these techniques in the contribution paragraph of the introduction, let us try to better detail them.
>
> The first idea is to use $1 \times 1$ convolutions to combine several random filters. In addition, we prune the random filters themselves rather than the tensor that combines them. In fact, a direct attempt leads to a polynomial increase: a straight-forward way of applying [[PRN+]]'s approach to convolution networks would be to prune the second layer. However, this results in trying to approximate a given filter as a sum of random filters. Such a generalization of subset sum sounds possible but would require $\Omega(d^2)$ random filters instead of a logarithmic number. A first non-trivial deviation from [[PRN+]] thus consists in applying pruning to the first layer in Lemma 1.
>
> A second one is to use Young’s inequality to bound the error propagation between layers while a more straight-forward analysis is possible in dense networks when both the input and weights vectors have $\ell_\infty$ norm at most 1. Note that due to the reuse of weights in convolutions, errors propagate more widely and bounding the overall error is more challenging than in a fully connected network.
>
> A third non-trivial adaptation was to apply pruning also in the second layer when approximating several filters in Lemma 2. This becomes necessary when limiting the approximation of 2D convolutions using 2D-convolutions. Indeed, it would be easier to approximate a 2D convolution by using a 3D-convolution (the use of several filters being captured by stacking several $U$ matrices along a third dimension as explained in the "Generalizations" paragraph in Section 2.1). However, appropriate pruning allows to encode the required convolution into a 2D convolution. Proving that we can approximate a 2D-convolutional network by pruning a larger 2D-convolutional network (rather than a larger 3D-convolutional network) is another difficulty that we managed to overcome.
>
>
> [PRN+]: https://arxiv.org/abs/2006.07990 "Pensia et al. (2020)"

---

### Public Comment · ~Wei_Zhang41 · 2021-11-12
**Questions for this paper**


The issues discussed in this paper are impressive and important. However, I suspect that the theorem proposed in this paper can prove the LTH in CNNs. There are some questions:

**1. What is the relationship between SLTH and LTH?**

The definition of SLTH in [2] is
>(SLTH 1) it stipulates that a network with random weights contains, with high probability, subnetworks that can approximate any given sufficiently smaller neural network.

The definition of SLTH in  [1] is
> (SLTH 2) sufficiently over-parameterized neural network with random initialization contains a subnetwork that achieves competitive accuracy (with respect to the large trained network), without any training.

The original LTH [3]:
>A randomly-initialized, dense neural network contains a subnetwork that is initialized such that—when trained in isolation—it can match the test accuracy of the original network after training for at most the same number of iterations.

How can we derive LTH from SLTH1 or SLTH2?  Malach et al 2020, Pensia et al 2020 and this paper claim that the issues they discussed in their paper are stronger than LTH, but none of these paper specifically explain why. Can the author provide details?
____

**2. How can we use Theorem 1 or informal version of Theorem 1 to prove LTH?**

The main theorem in this paper uses a $2\ell$ network to approximate a $1\ell$ network.  What is the relationship between the main theorem and the pruning? Shouldn't the pruned network and the original network have the same depth?

____

**3. About the distribution of the parameters?**

Many previous works [4][5][6] have found that the parameters do not follow a uniform distribution. Can the author provide the proof under the normal distribution?


----

----

----

[1] Malach et al, Proving the lottery ticket hypothesis: Pruning is all you need. ICML 2020.

[2] Pensia et al, Optimal lottery tickets via subset sum: Logarithmic over-parameterization is sufficient, NeurIPS 2020.

[3] Frankle et al, The lottery ticket hypothesis: Finding sparse, trainable neural networks, ICLR2019.

[4] Chen et al, AdderNet: Do We Really Need Multiplications in Deep Learning?, CVPR 2020.

[5] Huang et al, Rethinking the Pruning Criteria for Convolutional Neural Network. NeurIPS 2021.

[6] Liu et al, Rethinking the Value of Network Pruning, ICLR 2019.

---

> ### Author Response · Authors · 2021-11-13
> **Response to Questions**
>
> We are glad to receive questions from the community and we thank you for them. We believe they touch important points and that the answers can also be useful to the community in general.
> In the following we only quote the title of the questions to avoid repeating the further details provided in the original comment.
>
> #### 1. **What is the relationship between SLTH and LTH?**
>
> We believe the main hindrance to seeing the relationship between those statements is their informal form, especially regarding the notion of "training". This makes it hard to envisage a direct way for proving the original LTH; on the other hand, the SLTH offers the clever trick of trivializing training. In the original LTH, if the subnetwork already performs well at initialization, as long as training does not harm it, we are done. Thus, it is sufficient to notice that the SLTH (phrased as in *(SLTH 2)* in your comment, for instance) ensures that such subnetwork exists.
>
>
> #### 2. **How can we use Theorem 1 or informal version of Theorem 1 to prove LTH?**
>
> Reviewer *dycZ* raised a related question and [our response to their review](https://openreview.net/forum?id=Vjki79-619-&noteId=u52UfSBOb0j) (first answer) should also help answering this question.
> Our result (and previous ones) proves that sufficiently large randomly initialized networks can, with high probability, be pruned to yield a subnetwork that performs well even without training. We hope that the answer above clarifies that this implies that it is possible to prune neural networks at initialization and obtain subnetworks that can be successfully trained. Besides all of the empirical evidence for this (first brought by the original LTH paper), it was not clear that this was true for broad classes of networks.
>
>
> #### 3. **About the distribution of the parameters?**
>
> We kindly defer to the third answer in [our response to Reviewer *DkZV*](https://openreview.net/forum?id=Vjki79-619-&noteId=3NJImqv9A1l).
> Appendix C of our text provides some further details on this matter.
>
>
> *We hope our answers were helpful. Do not hesitate to ask for further clarification.*

---

> > ### Public Comment · ~Wei_Zhang41 · 2021-11-14
> > **Re: Response to Questions**
> >
> > I would like to thank the author for the feedback！
> >
> > **Q1. About the relationship between SLTH and LTH**
> >
> > Thank you for your clear reply.  In fact, I can understand the clever trick  and the notion of "training" . My question is actually about the target network (e.g. "any given sufficiently smaller neural network") you want to approximate. For example, if I want to use SLTH1 to derive LTH, I should let "any given sufficiently smaller neural network" be the origin network (i.e. a network with random weights in SLTH1) and the subnetwork can approximate the origin network, Right?  But, the origin network should be larger than the subnetwork. Is this contradictory？
> >
> >
> > **Q2. About a $2\ell$  network to approximate a $1\ell$ network**
> >
> > After reading the first answer for Reviewer dycZ's review. I think the authors did not answer the question about depth directly. In fact, in my opinion, it is very important for the issue discussed in your paper. The pruned network and the original network should have the same depth. If the conclusion in this paper is "a $2\ell$  subnetwork can approximate a $1\ell$ network", I do not think it is related to pruning and the title & contents in this paper **seem to be overclaimed**. Is there anything wrong with my understanding？
> >
> > It is undeniable that the author and  (Malach et al 2020) provided a beautiful proof of an approximation theorem, but not LTH or SLTH (I believe). I think it would be better if the author can modify their title and related overclaim content.
> >
> > Moreover, I think the most difficult part of proving LTH lies in the analysis on a network of a given depth and width.

---

> > > ### Author Response · Authors · 2021-11-15
> > > **Response to Questions**
> > >
> > > Thank you for your interest in our work.
> > > We believe that the LTH statement is too informal to approach it too rigorously in a meaningful way.
> > > For example, since it does not prescribe a precise bound on the sparsity of the subnetwork, one could consider as valid subnetworks those obtained by removing a single parameter.
> > > Even though a rigorous statement of the LTH as not been yet agreed upon,
> > > we believe that the theoretical works on the SLTH (ours included) are significant contributions w.r.t. the goal of the research around the LTH, which is to understand the mechanism of pruning and its relationship with training.
> > >
> > > Next, we provide our precise interpretation of the relationship between the SLTH and the LTH.
> > > For clarity, let us denote by $R_0$ a dense randomly initialized network with $2\ell$ layers, by $R_T$ its trained state, by $P$ a subnetwork obtained by pruning $R_0$, and by $T$ the target "sufficiently smaller" network with $\ell$ layers.
> > > The original LTH asserts that a subnetwork $P$ can be found so that training leads it to match the performance of $R_T$.
> > > Our results (and previous works) ensure that $P$ can be built to approximate any target network $T$; in the following, let us consider $T$ to be the best possible network for the task that $R_T$ is trained for.¹
> > > Under this light, your questions seem to boil down to comparing the performance of the *trained* $2\ell$-layered $R_T$ with that of the *best possible* $\ell$-layered $T$.
> > > For a large-enough $R$, $T$ can be assumed to be essentially optimal for the task under consideration.
> > > One can then see that the works proving the SLTH imply a fair interpretation of the LTH, where $R_0$ is assumed to be *large enough*. On the other hand, a rigorous statement of the LTH should not impose severe requirements on the size of $R_0$ in order to allow non-trivial bounds on the sparsity on the subnetwork $P$.
> > >
> > > ¹Notice that in both hypothesis $P$ is $2\ell$ layers deep.

---

> > > > ### Public Comment · ~Wei_Zhang41 · 2021-11-16
> > > > **Question for the proof**
> > > >
> > > > I would like to thank the author for the feedback again!
> > > >
> > > >
> > > >
> > > > **1.About "let us consider $T$ to be the best possible network for the task that $R_T$ is trained for"**
> > > >
> > > > In the response, the author says that "$T$ can be assumed to be essentially optimal for the task under consideration." Is it a reasonable assumption? In other words,  can **the best possible** $l$ layer network $T$ ("sufficiently smaller") matches the performance of $R_T$ (2$\ell$ layers)?  In fact, this assumption may be wrong and counter-intuitive. Can the author provide the theoretical evidences for reasonableness of this assumption?
> > > >
> > > >
> > > > I sincerely hope the author can try to validate this assumption. If the author can prove this assumption, the proof in this paper may be complete  and it will solve LTH in CNN thoroughly . Even if we have a weaker assumption that
> > > > >the best possible $l$ layer network $T$ ("sufficiently smaller") can match the performance of $R_T$ (2$\ell$ layers) with a high probability in some conditions,
> > > >
> > > > I believe it also provide a huge contribution to the pruning community.

---

> > > > > ### Author Response · Authors · 2021-11-16
> > > > > **Response to Questions**
> > > > >
> > > > > We thank you for helping us in clarifying aspects of the field that might cause confusion. We would like to further clarify our previous answer.
> > > > > By "let us consider $T$ to be the best possible network for the task that $R_T$ is trained for" we mean that, since we can approximate any small enough network $T$, we should reason in terms of the best possible one for the task of interest (among those that are sufficiently small).
> > > > > Furthermore, regarding the question of whether the best possible $T$ can match the performance of $R_T$, we can consider $T$ to be large enough to achieve 0 loss, which implies that $R_T$ cannot perform better.

---

> > > > > > ### Public Comment · ~Wei_Zhang41 · 2021-11-17
> > > > > > **Can we require the network $T$ is both "small enough" and "large enough"?**
> > > > > >
> > > > > > **Can we require the network $T$ is both "small enough" and "large enough"?**
> > > > > >
> > > > > > I can understanding that the author want to discuss $T$ which is large enough (achieve 0 loss) and small enough relative to $R_T$ (that is, $R_T$ is larger). But it may be an unreasonable requirement.
> > > > > >
> > > > > > Let we consider a origin network $R_0$ with a given M parameters, where M is large enough (In pruning, the original network should be given). According to Informal version of Theorem 1 in this paper, since $M = O(k\log(\frac{k\ell}{\min (\epsilon,\delta)}))$ is a constant, we can always select a given small $\epsilon$ to make the $k$ small enough.
> > > > > >
> > > > > > In other words, we can not guarantee the $T$ can be large enough to achieve 0 loss. From your theorem,  the number of parameters of $T$, i.e. $k$, depends on  $\ell, \epsilon, \delta$ and $M$. Is there anything wrong with my understanding？

---

> > > > > > > ### Author Response · Authors · 2021-11-21
> > > > > > > **Response to Questions**
> > > > > > >
> > > > > > > Thank you for the exchange.
> > > > > > >
> > > > > > > The notion of error used by the works in question is rather strong: the supremum of the absolute output error over the entire domain.
> > > > > > > It should not be confused with a weight approximation as it is not related to the network's structure, but rather to the task under consideration.
> > > > > > >
> > > > > > > With that in mind, given a task, one can choose $\varepsilon$ based on it and take $k$ to ensure $T$ is large enough to be optimal¹. Then it suffices to take any network with at least $O(k \log \frac{k\ell}{\min \\{\varepsilon, \delta\\}})$ parameters.
> > > > > > >
> > > > > > > As we mentioned, a notion of "large enough" is necessary to the LTH and we remark that such notion has to depend on the task.
> > > > > > >
> > > > > > > ¹The parameter $\delta$ is also independent from the architecture, and we can deal with $\ell$ similarly to how we did with $k$.

---

### Decision · Program_Chairs · 2022-01-20

**Decision:**

Accept (Poster)

**Comment:**

The paper presents interesting new results for pruning random convolutional networks to approximate a target function. It follows a recent line of work in the topic of pruning by learning. The results are novel, and the techniques interesting. There are some technical issues that are easy to fix within the camera ready timeline (see comments of reviewers below). I would also suggest refining the title of the paper: the lottery ticket hypothesis has an algorithmic component too, which clearly is not covered by existence results.